# Long-Term Nasal Growth after Primary Rhinoplasty for Bilateral Cleft Lip Nose Deformity: A Three-Dimensional Photogrammetric Study with Comparative Analysis

**DOI:** 10.3390/jcm8050602

**Published:** 2019-05-01

**Authors:** Hyung Joon Seo, Rafael Denadai, Lun-Jou Lo

**Affiliations:** Department of Plastic and Reconstructive Surgery and Craniofacial Research Center, Chang Gung Memorial Hospital, Chang Gung University, Taoyuan 333, Taiwan; payton100@hanmail.net (H.J.S.); denadai.rafael@hotmail.com (R.D.)

**Keywords:** primary cleft rhinoplasty, intermediate cleft rhinoplasty, cleft nose deformity, outcome, bilateral cleft lip nose deformity, three-dimensional analysis

## Abstract

Nasal deformity is associated with congenital cleft lip and palate. Primary rhinoplasty for reconstruction of the nasal deformity at the time of bilateral cleft lip repair is a controversial issue in cleft care due to traditional teaching concerning the potential impairment of nasal growth. This study assessed long-term nasal growth in patients with bilateral cleft lip and palate who underwent primary rhinoplasty by a single surgeon between 1995 and 2002 and reached skeletal maturity (*n* = 39; mean: 19 ± 2 years). Normal age-, gender-, and ethnicity-matched subjects (*n* = 52) were enrolled for comparative analyses. Three-dimensional nasal photogrammetric measurements (10 linear, 4 angular, 6 proportional, 1 surface area, and 1 volume parameter) were collected from patients with bilateral cleft lip and palate and normal subjects. Patients who underwent rhinoplasty presented with significantly (all *p* < 0.05) smaller nasal tip projection and nasal tip angles, but greater values for nasal dorsum length, nasal protrusion, alar width, columellar height, dome height, columellar angle, labiocolumellar angle, nasal tip height ratio, nasal index, alar width/intercanthal distance ratio, and alar width/mouth width ratio compared to normal subjects. There were no differences (all *p* > 0.05) in nasal height, tip/midline deviation, nasal dorsum angle, dome-to-columella ratio, columella height/alar width ratio, area surface, and volume parameters between the two groups. This study shows that primary rhinoplasty performed in patients with bilateral cleft lip and palate during infancy does not result in deficiency of the nasal dimensions relative to controls.

## 1. Introduction

Management of patients with bilateral cleft lip and palate (BCLP) requires collaboration from infancy to maturity between the multidisciplinary cleft team, patient, and parents [1,2,3]. Nasal deformity is typically associated with congenital cleft lip and palate [1,2,3]. The reconstruction of a bilateral cleft lip and nasal deformity is especially challenging owing to complex intrinsic characteristics including a short columella, a flattened nasal tip, and splayed nostril bases [1,2,3]. While various surgical strategies are available for repair of bilateral cleft lip and nasal deformity merit recognition for the achievement of outstanding results, the synchronous correction of the nasal deformity (primary rhinoplasty) at the time of lip repair has been recognized as a major advance in the management of patients with BCLP [4,5,6,7,8,9,10,11,12,13,14,15,16].

Interestingly, proponents of primary rhinoplasty have argued that this approach is essential for reaching an optimal outcome in bilateral cleft lip reconstruction [7,8,9,10,11,12,13,14,15,16]. On the other hand, opponents have based their opinions mainly on the traditional teaching that “primary rhinoplasty may potentially interfere with nasal growth” [17,18,19,20]. Studies showing the absence of nasal growth interference had methodological limitations (e.g., mixed sample sizes, age groups, and/or methods used for nasal measurements) that hindered universal acceptance of primary rhinoplasty within cleft centers worldwide [21,22,23,24,25,26,27]. It is therefore essential that further outcome analyses are conducted by implementing a well-delineated methodology based on accurate quantitative evaluation of nasal dimensions and comparisons between treated patients and normal subjects. Three-dimensional (3D) photogrammetric technology allows for fast image capture with good-resolution color representation and provides an accurate, reliable, and reproducible system for quantifying complex nose structure [28,29,30,31,32,33]. However, we are not aware of any 3D photogrammetric investigation assessing long-term nasal growth after primary rhinoplasty in patients with BCLP. In addition to enabling linear and angular measurements along the topographic contour of the nose [28,29,30,31,32,33], 3D photogrammetry offers the ability to evaluate the nasal surface area and perform volumetric analyses. This allows us to assess nasal growth-related data, with no need to expose the patient to ionizing radiation [33,34,35,36,37,38].

From this perspective, Dr. Noordhoff’s approach to primary rhinoplasty and subsequent modifications have been adopted at the Chang Gung Craniofacial Center over the past three decades [7,39]. This center has assessed different outcomes of primary rhinoplasty with or without an intermediate rhinoplasty procedure [40,41,42], but has not performed nasal growth evaluation in those patients who reached skeletal maturity. A senior surgeon (L.-J.L.) treated a subgroup of patients with BCLP using Noordhoff’s principles for bilateral cleft rhinoplasty between the late 1990s and early 2000s. As these patients recently reached skeletal maturity, the potential influence of rhinoplasty procedures on nasal growth can now be fully measured.

The purposes of this 3D photogrammetric study were to assess long-term nasal growth in patients with BCLP who underwent primary rhinoplasty, and to compare the 3D-based data to those in normal controls.

## 2. Patients and Methods

This was an observational retrospective study involving consecutive non-syndromic patients with complete BCLP who underwent primary rhinoplasty, with or without intermediate rhinoplasty, by a single surgeon (L.-J.L.) at the Chang Gung Craniofacial Center between 1995 and 2002. All included patients had reached skeletal maturity (defined as completing the growth spurt and showing no further increase in height) [43,44] at the time of data collection. Subjects with no history of facial trauma, facial surgery, or craniofacial deformity were enrolled as normal controls and matched for age, sex, and ethnicity. For comparisons between patients with BCLP and normal subjects, all patients who underwent rhinoplasty were combined as a single group, regardless of the approach used.

Demographic, clinical, and 3D nasal photogrammetric-based outcome data were collected. Patients were excluded if they were older than 4 months at the time of undergoing primary rhinoplasty, older than 6 years at the time of undergoing intermediate rhinoplasty, had associated syndromes, had a history of nasal surgery between the primary/intermediate rhinoplasty and data collection, did not undergo adequate 3D imaging, or did not complete the follow-up observation (<17 years). Patients who underwent orthognathic surgery were also excluded. The study was approved by the Institutional Review Board (20180309B0) and complied with the 1975 Declaration of Helsinki, as amended in 1983.

### 2.1. Primary Cleft Rhinoplasty

All primary cleft lip and nose surgical repairs were performed at 3–4 months of age using Noordhoff’s method with muscle-repositioning, banked-fork flap, and primary rhinoplasty [7,39]. Presurgical nasoalveolar molding (NAM) and a postsurgical nasal conformer were routinely used [41]. For primary rhinoplasty, a supraperichondrial dissection of the lower lateral cartilages was performed through the columella and alar bases using blunt tenotomy scissors. For further mobilization of the lower lateral cartilages, the attachments of the alar base at the pyriform rim were bilaterally released by extending the incision from the pyriform aperture upward, between the upper lateral cartilage and lower lateral cartilages, to the dorsal cartilage. Two types of sutures (between the dorsal and lower lateral cartilages and between the upper lateral cartilages and lower lateral cartilages) were placed in each alar dome to advance the lower lateral cartilages superiorly, resulting in widening of the intercartilaginous incision region. The buccal mucosal flap (designated the L flap) was then advanced into the midpoint of the intercartilaginous incision and sutured in place for reconstruction of the intentionally created intercartilaginous defect.

### 2.2. Intermediate Cleft Rhinoplasty

Intermediate rhinoplasty was performed at preschool age to address select clinical issues and patient/parental complaints, depending upon the magnitude of the nasal deformity (short columella, wide nose, and/or flat tip). Elongation of the columella, repositioning of the lower lateral cartilages, and improvement of tip projection were achieved. We implemented combined maneuvers, including the open tip approach with a columella incision extending intranasally or a nasal tip incision for V to Y advancement; advancement of the nasal floor tissue onto the columella; conchal or costal cartilage grafting into the medial crura; and cinch, intercrural, and transdomal sutures [40]. Each specific procedure was selected according to the surgeon’s preference and the patient’s diagnosis.

### 2.3. D Photogrammetric Analysis

Imaging data were acquired using the 3dMD system (3dMD LLC, Atlanta, GA, USA) under standard conditions including natural head position, relaxed facial musculature, and habitual dental occlusion. The system was calibrated before every capture process. Data sets were analyzed using 3dMD Vultus software (version 2.2, 3dMD Inc., Atlanta, GA, USA), with previously verified accuracy and reliability [45,46].

The facial and nasal anatomical landmarks (Figure 1) were defined according to original Farkas descriptions [47]. Reference planes and measurements (10 linear, 4 angular, 6 proportional, 1 surface area, and 1 volume parameter; Figure 2, Figure 3, Figure 4, Figure 5, Figure 6 and Figure 7; Table 1) were standardized based on previous nasal photogrammetric studies [28,29,30,31,32,33,34,35,36,37,38,43,48,49,50,51]. A standard reference frame (horizontal, coronal, and sagittal plane) was set up before all landmark identifications. The zoom and rotation tools were adopted to properly identify and set the landmarks on the 3D nasal surface. All photogrammetric measurements were collected in duplicate by two independent evaluators, with a 1-month interval between each measurement, and the mean was used for analysis.

### 2.4. Statistical Analysis

The data distribution was verified by the Kolmogorov–Smirnov test, and the parametric independent *t*-test or non-parametric Mann–Whitney U test was performed accordingly. Intra- and inter-examiner reliabilities were analyzed with intraclass correlation coefficients (ICCs) based on a two-way random-effects model. For all tests, *p* < 0.05 was considered statistically significant. All analyses were performed using SPSS Version 20.0 (IBM Corp., Armonk, NY, USA).

## 3. Results

We included 39 patients with complete BCLP who underwent primary rhinoplasty and reached skeletal maturity in our study. The patients with BCLP had undergone only primary rhinoplasty (*n* = 9, 23%) or primary plus intermediate rhinoplasty (*n* = 30, 77%), the female to male ratio was 0.86 (*n* = 18 and *n* = 21, respectively), and the mean age was 19.4 ± 2.2 years at the time of data collection. The 52 normal subjects were age-, gender-, and ethnicity-matched.

### 3.1. Intragroup Comparative Analyses

Among the patients with BCLP, we found that nasal tip height ratio was greater (*p* < 0.05) in those who underwent primary plus intermediate rhinoplasty than in those with primary rhinoplasty only. We found no significant differences in other measurements between those who underwent primary rhinoplasty only and those who underwent primary plus intermediate rhinoplasty (Table 2).

Among the patients with BCLP, we found that males had significantly (all *p* < 0.05) greater nasal bridge length, nasal height, nasal protrusion, alar width, nasal tip projection, columellar height, dome height, nasal surface area, and nasal volume than females. There were no significant differences in the remaining measurement parameters (Table 3).

### 3.2. Intergroup Comparative Analysis

Patients with BCLP presented with significantly (all *p* < 0.05) smaller nasal tip projection and nasal tip angle, but greater nasal dorsum length, nasal protrusion, alar width, columellar height, dome height, columellar angle, labiocolumellar angle, nasal tip height ratio, nasal index, alar width/intercanthal distance ratio, and alar width/mouth width ratio when compared to normal subjects. There were no significant differences in the remaining measurement parameters, including nasal height, tip/midline deviation, nasal dorsum angle, dome-to-columella ratio, columella height/alar width ratio, nasal area surface, and nasal volume (Figure 8; Table 4).

Among male subjects, patients with BCLP presented with significantly (all *p* < 0.05) greater nasal protrusion, alar width, columellar height, dome height, nasal dorsum angle, nasal tip angle, columellar-labial angle, nasal tip height ratio, nasal index, alar width/intercanthal distance ratio, alar width/mouth width ratio, columella height/alar width ratio, and nasal volume compared to normal subjects. We found that male patients with BCLP had significantly (all *p* < 0.05) smaller nasal bridge length, nasal tip projection, and columellar angle. There were no significant differences in the remaining parameters (Table 5).

Among female subjects, patients with BCLP presented with significantly (all *p* < 0.05) greater nasal protrusion, alar width, dome height, nasal tip angle, columellar-labial angle, nasal tip height ratio, alar width/intercanthal distance ratio, alar width/mouth width ratio, and nasal volume compared to normal subjects. We found that female patients with BCLP had significantly (all *p* < 0.05) smaller nasal bridge length, nasal tip projection, and columellar angle. We found no significant differences in the remaining parameters (Table 6).

The intra-and inter-examiner reliabilities were excellent (ICC = 0.80–0.96) for all nasal measurements.

## 4. Discussion

From a historical point of view, the reconstruction of nasal deformities in those with BCLP during the growth period has been considered unacceptable because of concern that early nasal repair may interfere with future nasal growth. This is exemplified by what Aufricht said in 1955: “please do not touch the nasal tip until the child is at least a teenager” [17]. This dictum has been clinically challenged over time by the benefits of rhinoplasty performed at infancy, including improvement of nasal symmetry, balance, and aesthetics, and by attenuation of the patient/parental psychosocial burden [7,8,9,10,11,12,13,14,15,16,21,22,23,24,25,26,27]. However, a recent survey demonstrated that one-half of cleft surgeons continue to deny these potential benefits to their patients [20], mainly because of the hypothetical disturbance in nasal growth, which remains an open question in cleft care [17,18,19].

Most previous studies attempting to clarify this issue have limited nasal measurements to childhood and teenage years [17,18,19,21,23,24,25,27]. Furthermore, investigations reporting nasal growth data at maturity have included only a limited number of patients and the measurements have solely been based on the subjective opinion of the treating surgeon or two-dimensional photogrammetry of a single parameter (columellar–labial angle) [22,26]. In our study, we adopted a detailed quantitative evaluation of the nose from different perspectives, including measurements of full-nose structure (area surface and volume), and linear, angular, and proportional dimensions within and between the nasal subunits (dorsum, tip, and columella). In order to assess the two approaches to rhinoplasty (primary rhinoplasty and primary plus intermediate rhinoplasty), we evaluated the data distribution by sex, and compared both approaches to matched normal subjects. For this purpose, a comprehensive 3D nasal photogrammetric analysis was performed, based on prior studies that reported nasal growth in a normal healthy population and evaluated different endpoints in cleft lip and nose repair [28,29,30,31,32,33,34,35,36,37,38,43,48,49,50,51]. Our excellent intra-and inter-evaluator reliability scores support the rigorous validation process previously performed for the 3D photogrammetric system [45,46]. As the same reference planes and parameter definitions were consistently used for all measurements, it was expected that intrinsic errors associated with the 3D system would have been similar in all included subjects, with no or minimum interference with the intragroup and intergroup comparisons. Additionally, as 3D data were collected from healthy, normal individuals matched for sex, age, and ethnic group, valid deductions could be made from these comparisons.

Different parameters in the literature (e.g., nasal protrusion, nasal height, columellar-labial angle, and nasal volume) have been defined as relevant in assessing nasal growth in normal subjects and patients with cleft lip and palate [24,25,26,27,34,35,36,37,38,43]. Using exactly the same definitions for nasal growth-related parameters [24,25,26,27,34,35,36,37,38,43], this study showed that primary rhinoplasty does not impair linear, angular, and proportional dimensions of the nose in patients with complete BCLP who reach maturity. This study also showed that mature patients with BCLP who have undergone primary rhinoplasty had nasal surface area and nasal volume similar to those of normal subjects. These two particular parameters are extremely important for nasal growth investigations, as they indicate that significant differences in linear, angular, and proportional parameters are not associated with nasal growth disturbance, but rather with the trajectory of growth causing residual nasal deformity at maturity.

Based on our findings, patients with BCLP presented with four main clinical differences compared to normal subjects, including a longer columella (statistically revealed by a greater columellar height and similar dome-to-columella ratio), cephalic rotation of the nasal tip (smaller columella angle and nasal bridge length and greater columellar–labial angle, nasal protrusion, and nasal tip height ratio), insufficient nasal tip projection (smaller nasal tip projection and greater nasal tip angle), and greater alar parameters (greater alar width, alar width/intercanthal distance ratio, alar width/mouth width ratio, and nasal index). Overall, these characteristics are consistent with the typical clinical manifestations of bilateral cleft lip nasal deformity. While the adopted techniques (presurgical NAM, primary rhinoplasty with or without intermediate surgery, and use of a postsurgical nasal retainer) were successful in correcting the typical short columella, the other features were not addressed properly. Unsatisfactory clinical results were found in previous studies [8,14,24,25,26,27]: caudal attachment of the columella base to the premaxilla and scarring process with downward drift of the columella base may explain the cephalic rotation of the nasal tip; inadequate repositioning of lower lateral cartilages during the primary and intermediate rhinoplasty procedures may result in insufficient nasal tip projection and greater alar width; and the greater alar width feature may be further justified by the presence of a narrow mouth width as a consequence of transposition of lateral lip elements below the prolabium during primary repair of bilateral cleft lip deformity.

Despite the benefits of primary rhinoplasty for nasal reconstruction [7,8,9,10,11,12,13,14,15,16,21,22,23,24,25,26,27,39], some patients may require additional nasal surgical intervention at preschool age (intermediate rhinoplasty) to address a residual tip-related deformity, as this is the most critical period of psychosocial development [40,52,53,54]. If nasal growth disturbance after primary rhinoplasty is secondary to development of heavy scar tissue, as theorized in traditional teaching, patients with two rhinoplasties within the first 6 years of life would have an even greater nasal growth restriction when they reached maturity. However, we found no significant differences for all but one parameter when patients with only primary rhinoplasty were compared with those who underwent primary plus intermediate rhinoplasty. This further supports that primary rhinoplasty is not a harmful procedure, even in patients with cleft lip and palate who undergo two nasal tip surgeries during the growth period.

The evolution of bilateral cleft lip nasal management has led to a group of patients who have undergone primary plus intermediate rhinoplasty. The included patients were treated when presurgical NAM was initiated at our center [39,40,41,42]. While some of the unsatisfactory results may be explained by failure of NAM to achieve narrowing of the cleft width, the closed primary rhinoplasty technique also failed to attain satisfactory nasal tip reconstruction. Moreover, intermediate rhinoplasty may not have completely resolved residual nasal deformity after unsuccessful primary rhinoplasty, as demonstrated by the current data. We followed up our patients with BCLP over time and our surgical protocol evolved to address each new challenge. All technical modifications implemented over the past 20 years represent a continuous effort to provide our patients with a more predictable long-term outcome. Therefore, the protocol used when patients in this study were first treated differed from the present protocol. Currently, patients with BCLP are treated with a modified NAM approach and a semi-open primary rhinoplasty technique, with overcorrection using the Tajima reverse-U incision for release of interdomal fibrofatty tissue and proper repositioning of the lower lateral cartilages. Preliminary clinical results have demonstrated almost no need for intermediate rhinoplasty, but a study with long-term follow-up is still needed to objectively quantify the intermediate rhinoplasty rate and to monitor the effect of the current approach on nasal growth at maturity.

Based on the current and previous findings [17,18,19,21,22,23,24,25,26,27], primary rhinoplasty with or without an intermediate procedure should be advocated as a routine intervention for complete BCLP reconstruction, with no risk of potential compromise of normal nasal dimensions. The “no-touch” approach to the nose should be replaced by primary rhinoplasty performed by surgeons devoted to cleft surgery, using a meticulous technique in accordance with published principles [1,2,3]. This paradigm shift in cleft care definitely depends on further scientific evidence from future outcome studies by other senior surgeons as well as the reevaluation of multi-center studies conducted two decades ago [55]. Meanwhile, young surgeons can interpret and apply the present results in their cleft practice, as this study is based on bilateral cleft lip nasal reconstructions performed by a single surgeon in his first years of practice.

Some inherent limitations to this retrospective study design may have affected our results and must be considered when interpreting our findings. Our center practices primary rhinoplasty exclusively, and thus patients without primary rhinoplasty were not included in our study [39,40,41,42]. Additionally, we did not include different techniques for primary and intermediate bilateral cleft lip nasal reconstruction for additional comparisons, but we did select normal subjects as our controls, as we assumed that the endpoint of bilateral cleft lip nasal reconstruction should be normal nasal morphology. While a normal nose is theoretically possible, it is not always achievable in all patients. Multiple factors other than growth disturbance, including the severity of initial deformity and the limitations of the repair technique, may influence the final nasal outcome at maturity. Therefore, nasal repair after completion of craniofacial growth (secondary rhinoplasty) is frequently needed to improve both aesthetic and functional aspects in patients with BCLP [52,53,54]. Thus, our observations on residual nasal deformities may be useful for preoperative planning, as specific technical maneuvers can be used to address each type of deformity. Although we did not control for occlusal status, patients underwent standard cleft palate repair (two-flap palatoplasty) and secondary alveolar reconstruction with iliac bone grafting [56] to attenuate its influence in comparative analysis between primary rhinoplasty only and primary plus intermediate rhinoplasty approaches.

Despite these limitations, our 3D-based findings provide valuable information to guide the decision-making process of cleft teams and to set realistic parental expectations before primary rhinoplasty by predicting nasal growth and the status of residual deformities at maturity.

## 5. Conclusions

Primary rhinoplasty at 3–4 months of age, with or without intermediate rhinoplasty at preschool age, does not result in deficiency of the nasal dimensions in patients with complete BCLP relative to controls.

## Figures and Tables

**Figure 1 jcm-08-00602-f001:**
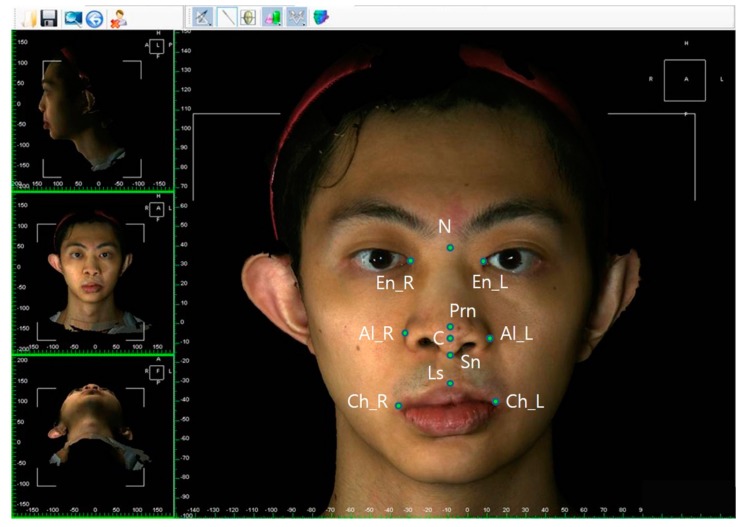
Three-dimensional imaging of a patient with bilateral cleft lip nose deformity at maturity showing anatomical landmarks: R, right; L, left; N, nasion; En, endocanthion; Prn, pronasale; C, columellar constructed point; Sn, subnasale; Al, alare; Ls, labiale superius; Ch, cheilion.

**Figure 2 jcm-08-00602-f002:**
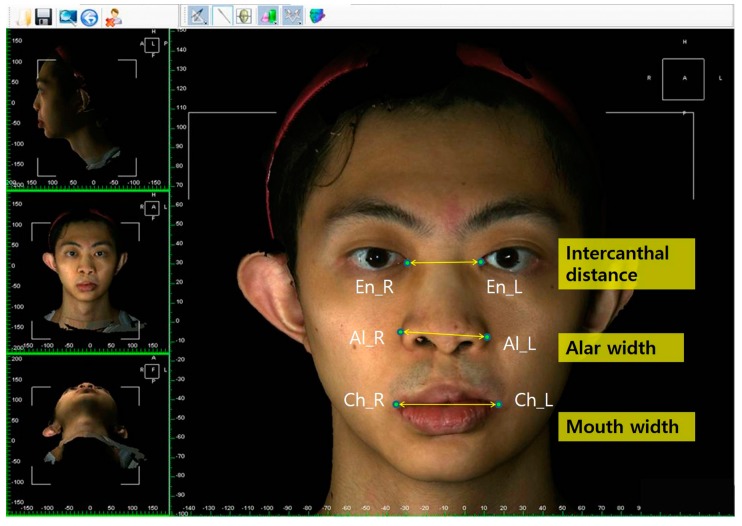
Representation of intercanthal distance, alar width, and mouth width.

**Figure 3 jcm-08-00602-f003:**
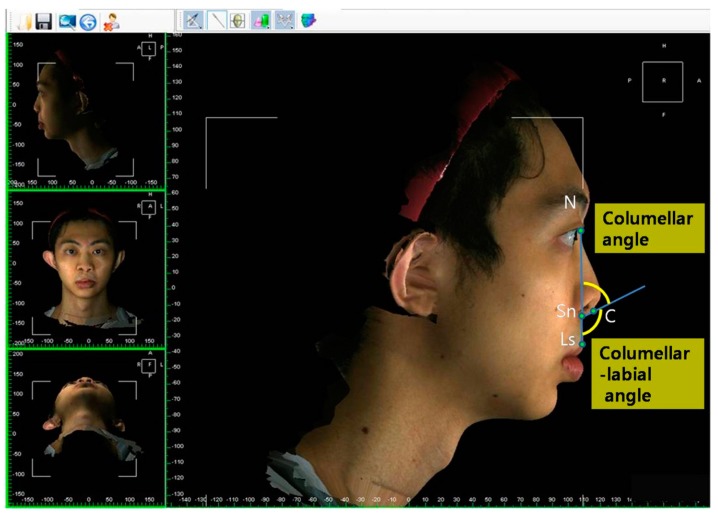
Representation of columellar angle and columellar-labial angle measurements. Changes or differences in these parameters are associated with cephalic or caudal rotation of the nasal tip.

**Figure 4 jcm-08-00602-f004:**
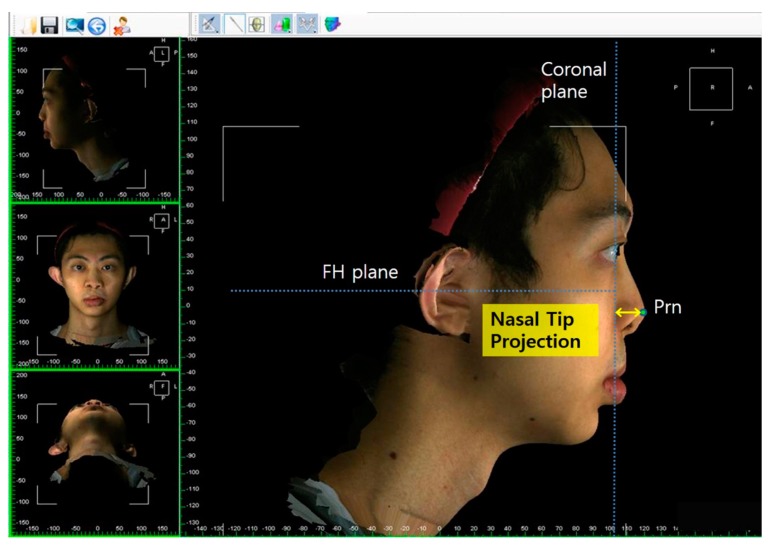
Representation of nasal tip projection.

**Figure 5 jcm-08-00602-f005:**
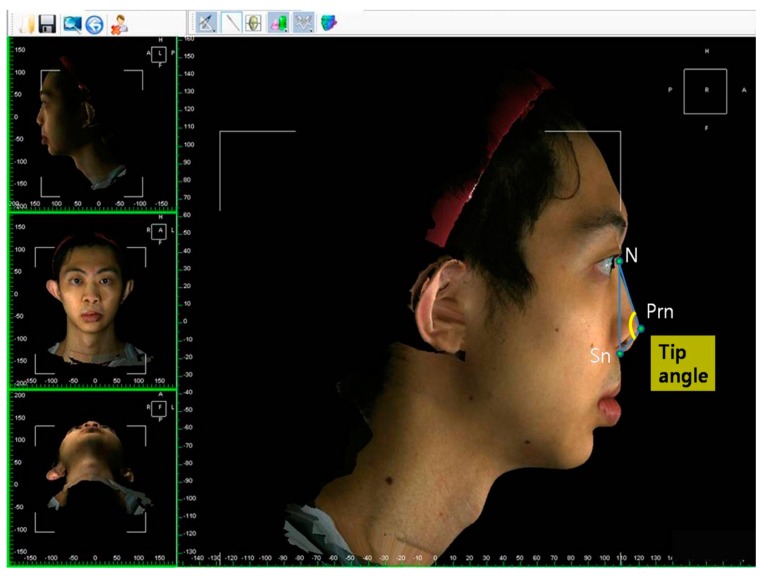
Representation of tip angle.

**Figure 6 jcm-08-00602-f006:**
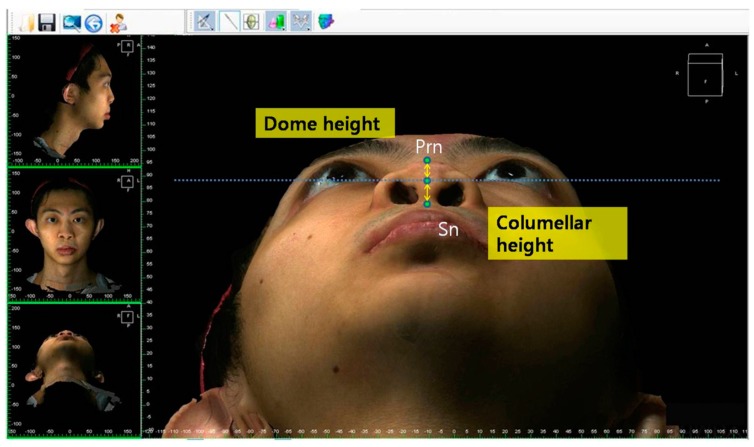
Representation of dome height and columellar height.

**Figure 7 jcm-08-00602-f007:**
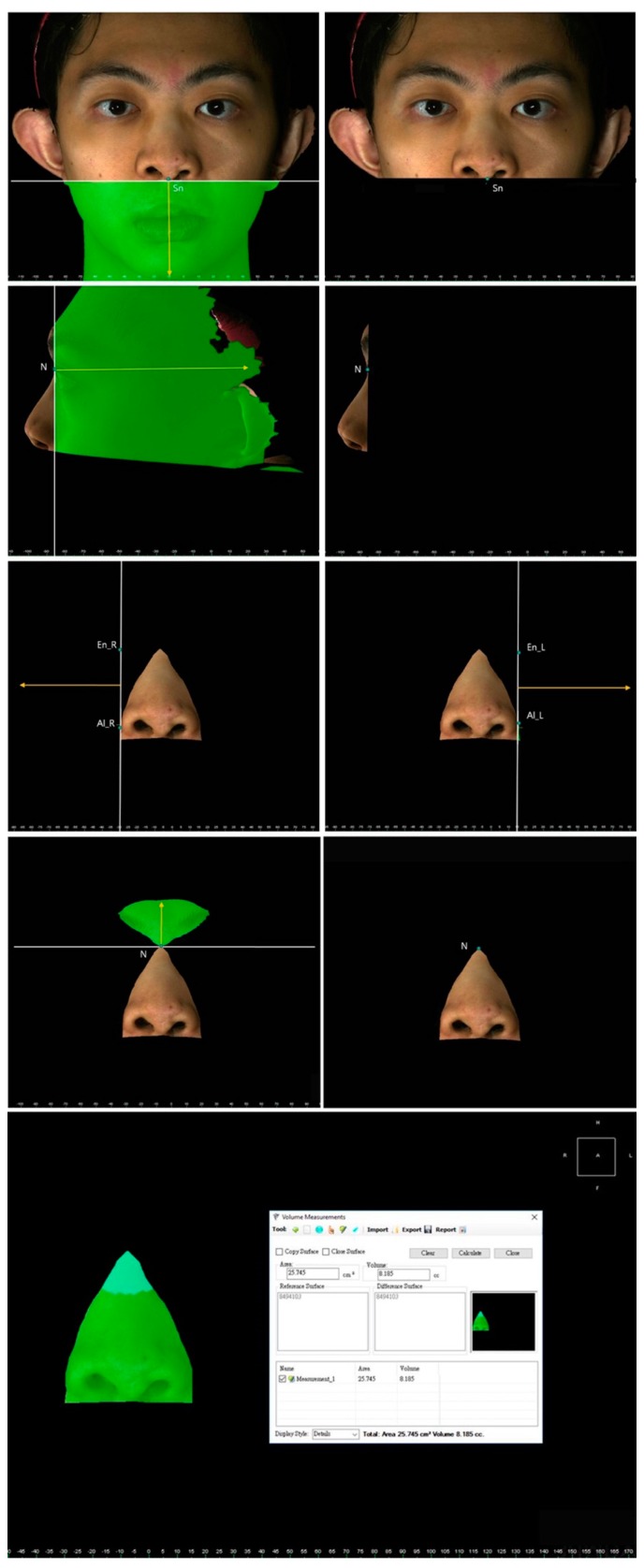
Representation of nasal surface area and nasal volume measurements. The nose was defined as a central three-dimensional object and regions without interest were trimmed from the inferior border of the glabella to the columellar–labial junction, and from the ala and junction of the cheek and nasal sidewall.

**Figure 8 jcm-08-00602-f008:**
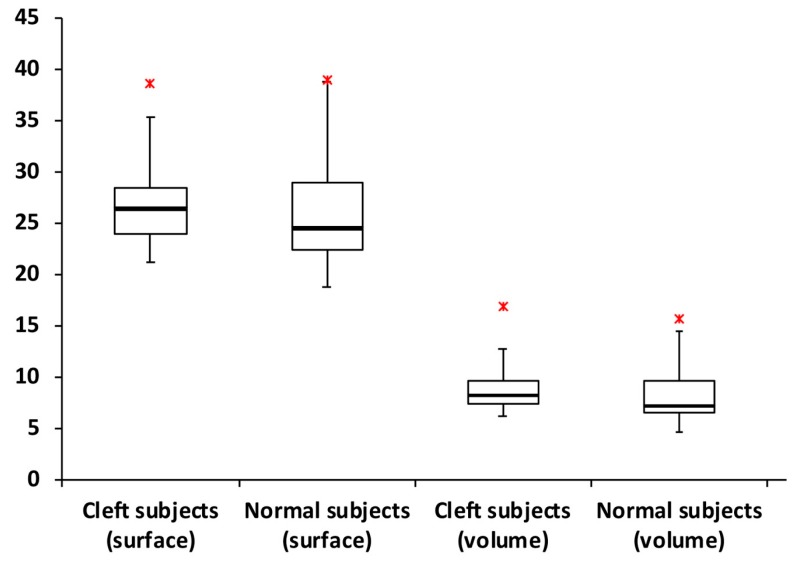
Box plot demonstrating the distribution of values for nasal area surface and nasal volume measurements. No significant differences were observed for comparisons between patients with bilateral cleft lip and palate and normal subjects. Red asterisks indicate maximum outliers’ values (*n* = 1 for each group).

**Table 1 jcm-08-00602-t001:** Parameters adopted for three-dimensional nasal photogrammetric analysis.

Parameters	Definitions
Landmarks	
Nasion (N)	Most depressed midline point superior to the nasal bridge
Pronasale (Prn)	Most anterior midpoint of the nasal tip
Subnasale (Sn)	Midpoint on the nasolabial soft tissue contour between the columella crest and the upper lip
Columellar constructed point (C)	Breakpoint at the end of the tangential line drawn from the Sn along the lower part of columella
Alare (Al)	Most lateral point on each alar contour
Endocanthion (En)	Soft tissue point located at the inner commissure of each eye fissure
Exocanthion (Ex)	Soft tissue point located at the outer commissure of each eye fissure
Labial superius (Ls)	Midpoint of the vermilion line of the upper lip
Cheilion (Ch)	Point located at each labial commissure
Tragion (T)	Soft tissue point located at the upper margin of each tragus
**Reference planes**	
T–Ex plane	Line passing through the T and Ex points
Soft tissue Frankfurt–Horizontal (FH) plane	Line passing through the T point and 17.6 degrees below the Ex–T plane
**Linear measurements**	
Nasal bridge length (NL)	Linear distance between the N and Prn points
Nasal height (NH)	Linear distance between the N and Sn points
Nasal protrusion (NP)	Linear distance between the Sn and Prn points
Alar width (Al–Al)	Linear distance between the right Al and left Al points
Nasal tip projection (TP)	Linear distance from coronal plane intersecting the alar facial groove and perpendicular to the FH plane to the Prn point
Tip/midline deviation (TD)	Linear distance from sagittal plane intersecting the N point and perpendicular to the FH plane to the Prn point
Columellar height (CH)	Linear distance between the midpoint of each nostril’s highest point and the Sn point
Dome height (DH)	Linear distance between the midpoint of each nostril’s highest point and the Prn point
Intercanthal distance (En–En)	Linear distance between the right En and left En points
Mouth width (Ch–Ch)	Linear distance between the right Ch and left Ch points
**Angular measurements**	
Nasal dorsum angle	Angulation calculated from intersecting lines drawn from the N to Sn points and from the N to Prn points (Sn–N–Prn)
Nasal tip angle	Angulation calculated from intersecting lines drawn from the N to Prn points and from the Sn to Prn points (N–Prn–Sn)
Columellar angle	Angulation calculated from intersecting lines drawn from the N to Sn points and from the Sn to C points (N–Sn–C)
Columellar–labial angle	Angulation calculated from intersecting lines drawn from the Sn to C points and from the Sn to Ls points (C–Sn–Ls)
**Proportional measurements**	
Nasal tip height ratio	Ratio between the distance from Sn to axial plane intersecting the Prn point and parallel to the FH plane and the distance from the N point to the same axial plane
Nasal index	Ratio between Al–Al and NH multiplied by 100
Alar width/intercanthal distance ratio	Ratio between Al–Al and En–En
Alar width/mouth width ratio	Ratio between Al–Al and Ch–Ch
Dome-to-columella ratio	Ratio between DH and CH
Columella height/alar width ratio	Ratio between CH and Al–Al

**Table 2 jcm-08-00602-t002:** Intragroup comparative analysis between primary rhinoplasty and primary plus intermediate rhinoplasty.

Measurements	Primary Rhinoplasty (*n* = 9)	Primary Plus Intermediate Rhinoplasty (*n* = 30)	*p*
Linear (mm)			
Nasal bridge length	40.72 ± 3.17	39.07 ± 3.49	0.243
Nasal height	51.50 ± 4.03	50.91 ± 4.31	0.777
Nasal protrusion	20.38 ± 4.22	20.40 ± 3.19	0.527
Alar width	42.29 ± 3.48	42.36 ± 3.32	0.764
Nasal tip projection	19.92 ± 4.37	19.98 ± 3.04	0.594
Tip/midline deviation	1.94 ± 1.33	1.38 ± 1.27	0.152
Columellar height	9.57 ± 2.08	9.17 ± 2.02	0.803
Dome height	11.75 ± 2.85	11.92 ± 2.13	0.571
**Angular (degrees)**			
Nasal dorsum angle	21.49 ± 3.91	21.36 ± 3.77	0.764
Nasal tip angle	111.23 ± 6.26	114.32 ± 7.22	0.334
Columellar angle	64.66 ± 7.02	62.27 ± 10.32	0.110
Columellar–labial angle	113.17 ± 21.29	119.00 ± 13.92	0.463
**Proportional**			
Nasal tip height ratio	0.34 ± 0.52	0.40 ± 0.08	0.030
Nasal index	82.23 ± 5.22	83.85 ± 10.20	0.790
Alar width/intercanthal distance ratio	1.10 ± 0.10	1.16 ± 0.14	0.205
Alar width/mouth width ratio	0.97 ± 0.11	0.96 ± 0.07	0.960
Dome-to-columella ratio	1.23 ± 0.19	1.35 ± 0.34	0.414
Columella height/alar width ratio	0.22 ± 0.03	0.22 ± 0.05	0.714
**Nasal surface area (mm^2^)**	27.53 ± 6.72	26.89 ± 3.76	0.689
**Nasal volume (mm^3^)**	9.57 ± 3.95	8.82 ± 1.74	0.594

**Table 3 jcm-08-00602-t003:** Intragroup comparative analysis between male and female subjects in patients with bilateral cleft lip and palate.

Measurements	Male Patients (*n* = 21)	Female Patients (*n* = 18)	*p*
Linear (mm)			
Nasal bridge length	40.75 ± 3.34	37.94 ± 3.00	0.009
Nasal height	52.92 ± 3.58	48.86 ± 3.87	0.002
Nasal protrusion	21.81 ± 3.64	18.73 ± 2.18	0.003
Alar width	44.15 ± 2.61	40.24 ± 2.80	<0.001
Nasal tip projection	21.80 ± 43.40	17.82 ± 1.45	<0.001
Tip/midline deviation	1.20 ± 1.24	1.87 ± 1.41	0.105
Columellar height	9.87 ± 2.29	8.56 ± 1.39	0.035
Dome height	12.81 ± 2.40	10.80 ± 1.57	0.003
**Angular (°)**			
Nasal dorsum angle	22.34 ± 4.14	20.29 ± 2.98	0.082
Nasal tip angle	112.55 ± 6.78	114.84 ± 7.36.	0.320
Columellar angle	65.17 ± 8.74	60.09 ± 10.13	0.101
Columellar–labial angle	115.38 ± 18.86	120.30 ± 11.19	0.322
**Proportional**			
Nasal tip height ratio	0.39 ± 0.07	0.38 ± 0.08	0.694
Nasal index	83.74 ± 7.12	83.16 ± 11.46	0.853
Alar width/intercanthal distance ratio	1.14 ± 0.12	1.15 ± 0.15	0.892
Alar width/mouth width ratio	0.97 ± 0.10	0.97 ± 0.06	0.926
Dome-to-columella ratio	1.36 ± 0.35	1.29 ± 0.26	0.501
Columella height/alar width ratio	0.22 ± 0.06	0.21 ± 0.04	0.491
**Nasal surface area (mm^2^)**	29.42 ± 4.10	24.25 ± 3.25	<0.001
**Nasal volume (mm^3^)**	10.40 ± 2.37	7.36 ± 0.93	<0.001

**Table 4 jcm-08-00602-t004:** Intergroup comparative analysis between patients with bilateral cleft lip and palate and normal subjects.

Measurements	Cleft Subjects (*n* = 39)	Normal Subjects (*n* = 52)	*p*
Linear (mm)			
Nasal bridge length	39.45 ± 3.45	42.19 ± 3.71	0.001
Nasal height	51.05 ± 4.20	49.60 ± 3.78	0.088
Nasal protrusion	20.39 ± 3.39	17.79 ± 1.63	<0.001
Alar width	42.34 ± 3.25	38.83 ± 3.03	<0.001
Nasal tip projection	20.0 ± 3.40	22.98 ± 3.20	<0.001
Tip/midline deviation	1.51 ± 1.29	1.21 ± 0.94	0.209
Columellar height	9.26 ± 2.01	7.98 ± 1.58	0.001
Dome height	11.88 ± 2.27	10.47 ± 1.91	0.002
**Angular (°)**			
Nasal dorsum angle	21.39 ± 3.75	20.32 ± 2.09	0.115
Nasal tip angle	113.61 ± 7.05	104.27 ± 5.02	<0.001
Columellar angle	62.82 ± 9.63	72.56 ± 7.46	<0.001
Columellar-labial angle	117.65 ± 15.79	99.99 ± 10.35	<0.001
**Proportional**			
Nasal tip height ratio	0.39 ± 0.07	0.31 ± 0.45	<0.001
Nasal index	83.47 ± 9.25	78.61 ± 7.37	0.009
Alar width/intercanthal distance ratio	1.15 ± 0.13	1.06 ± 0.10	0.001
Alar width/mouth width ratio	0.97 ± 0.08	0.82 ± 0.06	<0.001
Dome-to-columella ratio	1.32 ± 0.31	1.39 ± 0.46	0.459
Columella height/alar width ratio	0.22 ± 0.05	0.21 ± 0.05	0.216
**Nasal surface area (mm^2^)**	27.04 ± 4.52	25.88 ± 4.71	0.241
**Nasal volume (mm^3^)**	8.99 ± 2.39	8.06 ± 2.31	0.062

**Table 5 jcm-08-00602-t005:** Intergroup comparative analysis for male subjects.

Measurements	Cleft Subjects (*n* = 21)	Normal Subjects (*n* = 25)	*p*
Linear (mm)			
Nasal bridge length	40.74 ± 3.34	44.40 ± 3.01	<0.001
Nasal height	52.92 ± 3.60	51.87 ± 2.91	0.277
Nasal protrusion	21.81 ± 3.64	18.34 ± 1.66	<0.001
Alar width	44.15 ± 2.61	40.43 ± 2.69	<0.001
Nasal tip projection	21.80 ± 3.40	25.42 ± 2.40	<0.001
Tip/midline deviation	1.20 ± 1.12	0.97 ± 0.84	0.444
Columellar height	9.87 ± 2.29	7.72 ± 1.51	<0.001
Dome height	12.81 ± 2.40	11.31 ± 1.66	0.021
**Angular (°)**			
Nasal dorsum angle	22.34 ± 4.14	19.93 ± 1.78	0.012
Nasal tip angle	112.55 ± 6.78	104.00 ± 5.41	<0.001
Columellar angle	65.17 ± 8.74	74.47 ± 8.11	0.001
Columellar–labial angle	115.38 ± 18.86	98.41 ± 9.01	0.001
**Proportional**			
Nasal tip height ratio	0.39 ± 0.72	0.30 ± 0.03	<0.001
Nasal index	83.74 ± 7.12	78.05 ± 5.16	0.003
Alar width/intercanthal distance ratio	1.14 ± 0.12	1.08 ± 0.10	0.044
Alar width/mouth width ratio	0.97 ± 0.10	0.83 ± 0.07	<0.001
Dome-to-columella ratio	1.36 ± 0.35	1.53 ± 0.44	0.142
Columella height/alar width ratio	0.22 ± 0.06	0.19 ± 0.04	0.024
**Nasal surface area (mm^2^)**	29.42 ± 4.10	29.33 ± 4.21	0.943
**Nasal volume (mm^3^)**	10.4 ± 2.37	9.79 ± 2.03	0.358

**Table 6 jcm-08-00602-t006:** Intergroup comparative analysis for female subjects.

Measurements	Cleft Subjects (*n* = 18)	Normal Subjects (*n* = 27)	*p*
Linear (mm)			
Nasal bridge length	37.94 ± 3.00	40.15 ± 3.12	0.023
Nasal height	48.86 ± 3.87	47.49 ± 3.26	0.209
Nasal protrusion	18.73 ± 2.18	17.28 ± 1.46	0.010
Alar width	40.24 ± 2.80	37.35 ± 2.56	0.001
Nasal tip projection	17.82 ± 1.45	20.73 ± 1.94	<0.001
Tip/midline deviation	1.87 ± 1.41	1.43 ± 1.00	0.227
Columellar height	8.56 ± 1.39	8.22 ± 1.64	0.474
Dome height	10.80 ± 1.57	9.69 ± 1.82	0.040
**Angular (°)**			
Nasal dorsum angle	20.29 ± 2.98	20.69 ± 2.31	0.617
Nasal tip angle	114.84 ± 7.36	104.53 ± 4.73	<0.001
Columellar angle	60.09 ± 10.13	70.79 ± 6.45	<0.001
Columellar-labial angle	120.30 ± 11.19	101.46 ± 11.42	<0.001
**Proportional**			
Nasal tip height ratio	0.38 ± 0.08	0.32 ± 0.05	0.004
Nasal index	83.16 ± 11.46	79.13 ± 9.03	0.195
Alar width/intercanthal distance ratio	1.15 ± 0.15	1.04 ± 0.10	0.013
Alar width/mouth width ratio	0.97 ± 0.06	0.81 ± 0.06	<0.001
Dome-to-columella ratio	1.29 ± 0.26	1.25 ± 0.44	0.726
Columella height/alar width ratio	0.21 ± 0.04	0.22 ± 0.05	0.600
**Nasal surface area (mm^2^)**	24.25 ± 3.25	22.68 ± 2.23	0.061
**Nasal volume (mm^3^)**	7.36 ± 0.93	6.45 ± 1.03	0.004

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
