# Peer review of "Long-Term Nasal Growth after Primary Rhinoplasty for Bilateral Cleft Lip Nose Deformity: A Three-Dimensional Photogrammetric Study with Comparative Analysis"

_jcm, 2019, doi:10.3390/jcm8050602_

Round 1
Reviewer 1 Report
This study is helpful in helping answer some practical concerns about nasal growth.
The main question was whether nasal growth is
affected by primary rhinoplasty in bilateral Cleft lip and palate
patients. It is relevant
and original and adds to the literature.
The article is well-written and well-referenced and the conclusions were supported by the evidence.
Author Response
April 27, 2019
Prof. Dr. Emmanuel Andrès
Editor-in-Chief, Journal of Clinical Medicine
Dear Prof. Dr. Andrès,
We would like to thank you and your reviewers for considering our manuscript entitled “Long-term nasal growth after primary rhinoplasty for bilateral cleft lip nose deformity: a three-dimensional photogrammetric study with comparative analysis”(jcm-492131), for publication in Otolaryngology Section of Journal of Clinical Medicine. We appreciate the thorough and thoughtful review and for all the insightful comments and suggestions, as the comments have significantly improved our manuscript. We have addressed, on a point-by-point basis, all of the comments made by the reviewers. All changes have been included (marked in red) in our revised submission.
The article was reviewed by a professional English editing company.
Reviewer 1#:
1. This study is helpful in helping answer some practical concerns about nasal growth. The main question was whether nasal growth is affected by primary rhinoplasty in bilateral Cleft lip and palate patients. It is relevant and original and adds to the literature. The article is well-written and well-referenced and the conclusions were supported by the evidence.
Answer: We greatly appreciate the feedback from the reviewer.
We can provide additional modifications and explanations at the request of the Reviewers and/or Editorial Board. Thank you very much for your consideration of our manuscript.
The authors
Reviewer 2 Report
The study compares nasal measurements at the age of skeletal maturity, in bilateral cleft-lip patients who underwent primary and/or secondary rhinoplasty, with measurements in a normal control group. They find that only nasal protrusion and nasal angle were significantly reduced in the BCLP groups whereas other measures were found to not differ significantly or to be generally greater in the cleft group.
In general the study is well constructed, and the long-term follow-up of these patients provides valuable information. The paper is mostly well-written. I do have some suggestions and comments for improvement below.
MAJOR COMMENTS
Conclusions about nasal growth
The authors make some fairly strong claims that primary rhinoplasty “does not restrict nasal growth”. I understand their intention, but the claim as it is stated is not supported by the results. They have not actually investigated growth (e.g. they do not have longitudinal measurements of the same individuals). Nor have they, as they do acknowledge in the discussion, compared treated and untreated cleft patients. So they cannot claim that within cleft patients there is no effect of rhinoplasty on growth.
I think their intention is to say that, among those treated, there appears to be no deficiency in the nasal dimensions measured relative to controls. I believe this still supports their clinical recommendations (although I am not a surgeon or medical doctor) and the general claim that rhinoplasty is not harmful.
Could the authors please comment more on the dimensions for which there was a deficiency observed (nasal tip projection and angle). Are these too small to be clinically important?
Individual variation
The group-level statistics broadly support the authors’ claims. However they may obscure some important individual variation. For example, are there any individual cases in the treated group that display particularly small nasal dimensions? Might there be some cases where rhinoplasty does indeed have an impact?
MINOR COMMENTS
Figure 1 and Figure 2 appear to be the same.
When reporting the ICC the variant that is used must be reported, otherwise it is not clear how it can be interpreted it. McGraw, Kenneth O., and Seok P. Wong. "Forming inferences about some intraclass correlation coefficients." Psychological methods 1.1 (1996): 30
Ln 167 – ‘the data distribution was verified’ -> the distribution of data were shown to not significantly deviate from a normal distribution.
In the results, are the p-values reported from the independent t-test or from the Mann-Whitney U tests?
Author Response
April 27, 2019
Prof. Dr. Emmanuel Andrès
Editor-in-Chief, Journal of Clinical Medicine
Dear Prof. Dr. Andrès,
We would like to thank you and your reviewers for considering our manuscript entitled “Long-term nasal growth after primary rhinoplasty for bilateral cleft lip nose deformity: a three-dimensional photogrammetric study with comparative analysis”(jcm-492131), for publication in Otolaryngology Section of Journal of Clinical Medicine. We appreciate the thorough and thoughtful review and for all the insightful comments and suggestions, as the comments have significantly improved our manuscript. We have addressed, on a point-by-point basis, all of the comments made by the reviewers. All changes have been included (marked in red) in our revised submission.
The article was reviewed by a professional English editing company.
Reviewer 2#:
1. The study compares nasal measurements at the age of skeletal maturity, in bilateral cleft-lip patients who underwent primary and/or secondary rhinoplasty, with measurements in a normal control group. They find that only nasal protrusion and nasal angle were significantly reduced in the BCLP groups whereas other measures were found to not differ significantly or to be generally greater in the cleft group. In general the study is well constructed, and the long-term follow-up of these patients provides valuable information. The paper is mostly well-written.
Answer: We greatly appreciate the feedback from the reviewer as the comments have significantly improved our manuscript.
2. The authors make some fairly strong claims that primary rhinoplasty “does not restrict nasal growth”. I understand their intention, but the claim as it is stated is not supported by the results. They have not actually investigated growth (e.g. they do not have longitudinal measurements of the same individuals). Nor have they, as they do acknowledge in the discussion, compared treated and untreated cleft patients. So they cannot claim that within cleft patients there is no effect of rhinoplasty on growth. I think their intention is to say that, among those treated, there appears to be no deficiency in the nasal dimensions measured relative to controls. I believe this still supports their clinical recommendations (although I am not a surgeon or medical doctor) and the general claim that rhinoplasty is not harmful.
Answer: We replaced “does not restrict nasal growth” by “no deficiency in the nasal dimensions” as requested. It was also included that primary rhinoplasty is not a harmful procedure. It was addressed in the abstract and body of text (discussion and conclusion sections).
3. Could the authors please comment more on the dimensions for which there was a deficiency observed (nasal tip projection and angle). Are these too small to be clinically important?
Answer: For nasal-related dimensions, small changes in linear distances and angles may positively or negatively impact the overall shape and aesthetic of nose, a complex 3D structure. Particularly, nasal tip projection and angle measurements present clinical relevance. Therefore, as our findings related to linear and angular measurement features have clinical repercussion, nasal repair after completion of craniofacial growth (secondary rhinoplasty after orthognathic surgery [if need]) is frequently needed to improve both esthetic and functional aspects in patients with bilateral cleft lip nose deformity. It was addressed in the discussion section, as requested:
- Discussion section: Our findings demonstrated that cleft subjects presented 4 main clinical differences compared to those in normal subjects, including a longer columella (statistically revealed by a greater columellar height and similar dome-to-columella ratio), cephalic rotation of the nasal tip (smaller columella angle and nasal bridge length; and greater columellar-labial angle, nasal protrusion, and nasal tip height ratio), insufficient nasal tip projection (smaller nasal tip projection and greater nasal tip angle), and a greater alar parameters (greater alar width, alar width/intercanthal distance ratio, alar width/mouth width ratio, and nasal index). Overall, these characteristics are consistent with the typical clinical manifestations of patients with bilateral cleft lip nasal deformity. These clinical results were similarly revealed in previous studies [8,14,24-27].
Therefore, nasal repair after completion of craniofacial growth (secondary rhinoplasty) is frequently needed to improve both esthetic and functional aspects in BCLP [52-54].
4. The group-level statistics broadly support the authors’ claims. However, they may obscure some important individual variation. For example, are there any individual cases in the treated group that display particularly small nasal dimensions? Might there be some cases where rhinoplasty does indeed have an impact?
Answer: Our study showed that mature cleft individuals who have undergone primary rhinoplasty had the nasal surface area and nasal volume similar to those in normal subjects. These two particular parameters (nasal surface area and nasal volume) are extremely important for nasal measurement-related investigations, as they indicate that significant differences in linear, angular, and proportional parameters are not associated with disturbance of nasal dimensions, but rather with the trajectory of growth causing residual nasal deformity at maturity. We created boxplot graphics (Figure 8) to illustrate the distribution of data (cleft and normal subjects) for volume and surface parameters. These graphics reveal no minimum outliers for cleft individuals included in this study.
5. Figure 1 and Figure 2 appear to be the same.
Answer: Figure 1 contains the anatomical landmarks adopted for nasal-related measurements, whereas Figure 2 contains the definitions of three linear measurements. It was reviewed, as requested.
6. When reporting the ICC the variant that is used must be reported, otherwise it is not clear how it can be interpreted it. McGraw, Kenneth O., and Seok P. Wong. "Forming inferences about some intraclass correlation coefficients." Psychological methods 1.1 (1996): 30
Answer: ICC estimates and their 95% confident intervals were calculated based on a 2-way random-effects model. It was addressed in the Methods section, as requested.
7. Ln 167 – ‘the data distribution was verified’ -> the distribution of data were shown to not significantly deviate from a normal distribution. In the results, are the p-values reported from the independent t-test or from the Mann-Whitney U tests?
Answer: All variables have the distribution verified by the Kolmogorov-Smirnov test, and parametric independent t-test or non-parametric Mann-Whitney U test (Wilcoxon rank-sum test) were performed accordingly. It was addressed in the Methods section, as requested.
We can provide additional modifications and explanations at the request of the Reviewers and/or Editorial Board. Thank you very much for your consideration of our manuscript.
The authors